# Aging-Resistant Behavior and Room Temperature Electron Spin Resonance of Nd^3+^ in Singly and Doubly Doped BaTiO_3_ Ceramics Associated with Preservation History

**DOI:** 10.3390/ma12030451

**Published:** 2019-02-01

**Authors:** Dayong Lu, Lv Ji, Junwei Liu

**Affiliations:** 1Key Laboratory for Special Functional Materials in Jilin Provincial Universities, Jilin Institute of Chemical Technology, Jilin 132022, China; jilv6212@163.com (L.J.); jwju@foxmail.com (J.L.); 2State Key Laboratory of Superhard Materials, Jilin University, Changchun 130012, China

**Keywords:** barium titanate ceramics, point defects, dielectric properties, aging resistant, electron spin resonance

## Abstract

The (Ba_0.96_Nd_0.04_)Ti_0.99_O_3_ (BN4T) and (Ba_0.96_Nd_0.04_)(Ti_0.94_Ce_0.05_)O_3_ (BN4TC5) ceramics were prepared via the mixed oxide route exhibited tetragonal and pseudo-cubic structures, respectively. After they were preserved over a long period of time, a broader electron spin resonance (ESR) signal at *g* = 2.338 and a narrow ESR signal at *g* = 2.151 were detected at room temperature (RT) for BN4T and BN4TC5, respectively. They most likely originated from the Nd^3+^ Kramers ions in BN4T and Nd^3+^-Ce^4+^ defect complexes in BN4TC5 ceramics, respectively. The origins of these two ESR signals and the aging-resistant dielectric behavior are further discussed.

## 1. Introduction

In the dielectric field, neodymium (Nd) acts as an effective dopant to be widely investigated in singly [1,2,3,4,5] and doubly doped BaTiO_3_ (ABO_3_) ceramics [6,7,8]. In most cases, Nd^3+^ substitutes dominate onto Ba sites [1,2,3,4,7]. In the particular case of Ba/Ti = 1 and long-term sintering at 1400 °C for 2–3 days, a self-compensated amphoteric behavior of Nd^3+^ in BaTiO_3_ was proposed by Hirose et al. [5]. In general, Nd^3+^ prefers Ba sites to Ti sites based on unit-cell volume (*V*_0_) calculations and ionic size comparison [1,2,3,4,5,6,7,8].

The powder electron spin resonance (ESR) technique is valid in detecting some rare-earth Kramers ions (with an odd number of electrons) in doped BaTiO_3_ ceramics at room temperature (RT) [9,10,11,12,13,14]. Except for our reports on ESR [9,10,11,12,13,14], recent studies on ESR for doped BaTiO_3_ are very rare and focused on the magnetic properties of ceramics [15]. Only ESR signals of three kinds of rare-earth ions with a half-filled 4*f* shell (4*f*^7^, ^8^S_7/2_)—metastable Ba-site Eu^2+^ (*g* = 1.98) [9], Ti-site Tb^4+^ (*g* = 5.50–6.58) [10,11,12], and Gd^3+^ (*g* = 1.985) [12]—can be repeatedly detected in doped BaTiO_3_ at RT, which is attributed to their longer spin-lattice relaxation time of the 4*f*^7^ electronic configuration at RT. A thermally accessible *g* = 3.77 signal associated with Ti-site Pr^4+^ (4*f*^1^, ^3^H_4_) was unexpectedly observed above 120 °C, which relates to the preservation history of the Pr-doped BaTiO_3_ sample [13]. A broad *g* = 2.23 signal associated with Er^3+^ (4*f*^11^, ^4^I_15/2_) Kramers ions in the self-compensation mode ${\mathrm{Er}}_{\mathrm{Ba}}^{•}-{\mathrm{Er}}_{\mathrm{Ti}}^{\prime }$ was observed at RT [14]. The defect notation adopted is that suggested by Kröger and Vink [8]. Other light and heavy rare-earth ions in doped BaTiO_3_ rarely exhibit an ESR response at RT.

It is known that the ESR of Nd^3+^ in BaTiO_3_ crystals can be observed in the rhombohedral phase at low temperatures (such as *T* = 7 K) using an X-band frequency; at *T* ≥ 10 K, the spectrum vanishes [16]. In other host lattices, such as Y_2_SiO_5_, the ESR of Nd^3+^ was also studied at low temperatures (*T* = 5–15 K) [17]. Recently, a narrow ESR signal at *g* = 2.151 was discovered when we re-investigated the ESR of high-permittivity and low-loss (Ba_1–*x*_Nd*_x_*)(Ti_1–*x*/4–*y*_Ce*_y_*)O_3_ ceramics prepared in 2006 [18]. For this finding, we further investigated the ESR of another (Ba_1–*x*_Nd*_x_*)Ti_1–*x*/4_O_3_ (*x* = 0.04) ceramic prepared in 2006 and found a different, broader ESR signal. In this work, we report these two ESR signals in singly and doubly doped BaTiO_3_ ceramics associated with preservation history. Their origins are discussed in association with the spin-lattice relaxation time of Nd^3+^ and the rhombohedral distortions existing in the aged powders.

Ferroelectric ceramics are subject to degradation either during electrical loading (fatigue) or with time in the absence of an external mechanical or electrical load (aging) [19]. Recent studies on aging behavior mainly focused on mechanisms of piezoelectric ceramics [19,20] and change in polarization-electric field (*P*–*E*) hysteresis loops [21]. However, the time-induced aging effect has prompted surprisingly little concern in the dielectric behavior. The high-permittivity (Ba_1−*x*_Nd*_x_*)(Ti_1−*x*/4−y_Ce*_y_*)O_3_ (*x* = 0.04) ceramic meets the Y5V specification (−82% ≤ (*ε*′−*ε*′_RT_)/ε′_RT_ ≤ +22% in the temperature range −30 to 85 °C) [18], making it ideal for decoupling applications within a limited temperature range. The investigation of the time-induced aging property of this ceramic and reports on its advantageous aging-resistant performance (a decrease in *ε*′_m_ by 6% after 12 years) are of great importance in the actual Y5V application.

## 2. Methods

### 2.1. Materials

The initial materials were reagent-grade BaCO_3_, TiO_2_, CeO_2_ (Kanto Chem., Tokyo, Japan), and Nd_2_O_3_ (Wako Pure Chem., Tokyo, Japan). The (Ba_0.96_Nd_0.04_)Ti_0.99_O_3_ (BN4T) and (Ba_0.96_Nd_0.04_)(Ti_0.94_Ce_0.05_)O_3_ (BN4TC5) ceramics were prepared in 2006 using the mixed oxide route, as described elsewhere [10]. The stoichiometric mixtures in accordance with the above metal ratios (Ba/Nd/Ti) for BN4T and (Ba/Nd/Ti/Ce) for BN4TC5 were carefully mixed. The mixtures were calcined in air at 1100 °C for 5 h to decarbonate the initial material BaCO_3_. The purpose of “decarbonating” in the calcination stage is to avoid the release of CO_2_ and to benefit ceramic densification in the following sintering stage. The calcined mixtures with polyvinyl alcohol (PVA) aqueous solution were pressed uniaxially at 200 MPa into 12.0 mm diameter pellets. These pellets were sintered at 1450 °C for 24 h in air, followed by furnace cooling to room temperature.

### 2.2. Characterization

Powder X-ray diffraction (XRD) data were collected with Cu Kα radiation at RT using a Rint 2200 X-ray diffractometer (Rigaku, Tokyo, Japan) in 2006 and using a DX-2700 X-ray diffractometer (Dandong Haoyuan, Dandong, China) in 2017. Crystal structures were determined by MS Modeling (Accelry Inc., San Diego, CA, USA) and Cu Kα_1_ radiation (λ = 1.540562 Å). A quantity of 30 mg of powder for each sample was taken and placed into a quartz tube for ESR measurements. The symbols and the preservation history of BN4T and BN4TC5 are shown in Table 1.

ESR spectra were measured at RT using a JES-RE3X spectrometer (JEOL, Tokyo, Japan) at the X-band frequency (9.148 GHz) in 2006 and using an A300-10/12 spectrometer (Bruker, Billerica, MA, USA) at 9.85 GHz in 2017. The *g*-factor of signals was calculated according to the relationship *hν*_0_ = *gβH*, where *h* is Planck’s constant (*h* = 6.626 × 10^−34^ J s), *ν*_0_ is the microwave frequency, *β* is the Bohr magnetron (*β* = 9.262 × 10^−24^ J/T), and *H* is the magnetic field strength. EasySpin software (http://easyspin.org/easyspin/documentation/references.html) was used to simulate the X-band ESR spectra of Nd^3+^ ions in BaTiO_3_. Polished disks 0.8 mm in thickness were used for electrical measurements. In 2006, the temperature dependence of the dielectric permittivity (*ε*′–*T*) was measured for BN4TC5 from −100 to 200 °C at 1 kHz, under an electric field of 100 V/cm and a heating rate of 2 °C/min using a combination of instrumentation (AD-3521 FFT Analyzer, A&D Co., Ltd., Tokyo, Japan); Temperature Control Unit Model SU 500, NEC; Digitizing Oscilloscope 5450A, Hewlett-Packard, Palo Alto, CA, USA). In 2017, the *ε*′–*T* curve was re-measured using a Concept 41 Dielectric/Impedance spectrometer (Novocontrol, Montabaur, Germany) with an applied voltage of 1 V and at 1 kHz. A Nd:YAG laser was used for excitation in obtaining the Raman spectra of the ceramics and photoluminescence of Nd^3+^ using a LabRAM XploRA Raman spectrometer (Horiba Jobin Yvon, Bensheim, Germany) with a 532 nm line focused on a spot approximately 2 μm in diameter. The laser power level was limited to 0.1% (Filter, Pawtucket, RI, USA) of the normal output of 25 mW. The accumulation time and resolution were 2 s and 2.7 cm^−1^, respectively.

## 3. Results

Powder XRD spectra of the BN4T and BN4TC5 ceramics were measured in 2006 and again in 2017, as shown in Figure 1. Miller indices are given for these two samples in Figure 1. BN4T-1 has a tetragonal perovskite structure with the space group P4*mm* (Figure 1b). Its lattice parameters are *a* = 3.9965 Å and *c* = 4.0106 Å; the unit-cell volume is *V*_0_ = 64.06 Å^3^, which is lower than that of the tetragonal BaTiO_3_ (*V*_0_ = 64.41 Å^3^, JCPDS Cards No. 5-626), suggesting the dominant occupations of Ba sites by Nd^3+^ [1,2,3]. BN4TC5-1 has a pseudo-cubic perovskite structure (the space group P*m*3*m*) with *a* = 4.0275 Å and *V*_0_ = 65.33 Å^3^ (Figure 1a). The peak splitting and the symmetric peak in the vicinity of 45° (Figure 1 insets) are characteristic of tetragonal and cubic symmetry, respectively. The *V*_0_ of BN4TC5-1 is far higher than that of BN4T-1, indicating that Nd and Ce ions mainly substituted on Ba sites for Nd^3+^ and Ti sites for Ce^4+^ [18]. The powder XRD spectra of BN4T-2 and BN4TC5-2 were re-measured in 2017. These two samples still maintained the initial tetragonal and pseudo-cubic perovskite structures. No evident phase transition was observed. The long-term preservation led to a slight expansion in *V*_0_ for BN4TC5-2 (*V*_0_ = 65.37 Å^3^) relative to BN4TC5-1 (*V*_0_ = 65.33 Å^3^).

The temperature dependences of the dielectric permittivity (*ε*′) and the dielectric loss (tan*δ*) for BN4TC5 were measured in 2006 and again in 2017, as shown in Figure 2. As time went by, the permittivity maximum (*ε*′_m_) decreased by 6% and the temperature of the permittivity maximum (*T*_m_) shifted towards a higher temperature from 35 °C in 2006 to 40 °C in 2017. The latter resulted in the invalidation of Y5V specification. Although BN4TC5 has good aging-resistant behavior in *ε*′, the room-temperature loss exhibited a slight increase but increased rapidly above RT.

The ESR spectra of BN4T measured in 2006 and 2017 are shown in Figure 3. A Mn^2+^ sextet signal, which is induced by the reduction from Mn^4+^/Mn^3+^ to Mn^2+^ impurities owing to the doping effect of Nd^3+^ (Equations (1) and (2)), appears in all BN4T samples, indicating that Mn^2+^ in BN4T in air cannot be oxidized to high-valence Mn ions after long-term preservation of the sample.
(1)$${\mathrm{Mn}}_{\mathrm{Ti}}^{\times }+2\;e^{\prime }\to {\mathrm{Mn}}_{\mathrm{Ti}}^{\prime \prime }$$
(2)$${\mathrm{Mn}}_{\mathrm{Ti}}^{\prime }+\;e^{\prime }\to {\mathrm{Mn}}_{\mathrm{Ti}}^{\prime \prime }$$


A *g* = 2.004 signal associated with ionized Ti vacancies (V_Ti_) (Equations (4) and (5)) [22,23] was detected for BN4T-1 prepared in 2006 because the dominant substitutions of Nd^3+^ onto Ba sites create Ti vacancies (Equation (3)).
(3)$${4\;\mathrm{Nd}}_{\mathrm{Ba}}^{•}\to V_{\mathrm{Ti}}^{\prime \prime \prime \prime }$$
(4)$$V_{\mathrm{Ti}}^{\prime \prime \prime \prime }+\;3e^{\prime }\to V_{\mathrm{Ti}}^{\prime }$$
(5)$$V_{\mathrm{Ti}}^{\prime \prime \prime \prime }+\;e^{\prime }\to V_{\mathrm{Ti}}^{\prime \prime \prime }$$


After the BN4T ceramic powder was stored in a desiccator for eleven years, a broader signal at *g* = 2.338 was unexpectedly observed for BN4T-3 at RT in 2017 (Figure 3b). This broader signal vanished at RT after the BN4T-2 ceramic powder was heat-treated at 500 °C (Figure 3c).

The ESR spectra of BN4TC5 measured in 2006 and 2017 are shown in Figure 4. The V_Ti_-related signal appears in all BN4TC5 samples (Equations (3)–(5)). The intensity of the Mn^2+^ sextet signal of BN4TC5 is far lower than that of BN4T, which suggests that the formation of a ${\mathrm{Nd}}_{\mathrm{Ba}}^{•}-{\mathrm{Ce}}_{\mathrm{Ti}}^{\prime \prime }$ complex in BN4TC5 [18] reduces the doping effect of ${\mathrm{Nd}}_{\mathrm{Ba}}^{•}$ in BN4T. Except for V_Ti_- and Mn^2+^-related signals, a narrow *g* = 2.151 signal with a linewidth of 26 Gs was observed in 2017 (Figure 4b). This signal vanished at RT after the BN4TC5-2 sample was heat-treated at 500 °C (Figure 4c).

## 4. Discussion

### 4.1. Probable Origin and Simulation of Nd^3+^ ESR in BN4T at RT

For BN4T, Ba^2+^, Ti^4+^, and O^2−^ have no ESR response; the signals from ionized Ba- (*g* = 2.004), Ti- (*g* = 1.974), and O-vacancies (*g* = 1.974) are unrelated to the broader signal at *g* = 2.338 in BN4T-2 (Figure 3b) [8,9,10,11,12,13,14,18,22,23,24,25]. It is therefore inferred without considering the spin-lattice relaxation time that this broader signal should originate from Nd^3+^ (3*f*^3^) in the tetragonal BN4T, without the possibility of other point defects.

The spin-Hamiltonian of the paramagnetic Nd^3+^ centers in the tetragonal BaTiO_3_, including the Zeeman and the hyperfine terms, can be written as
(6)$$\hat{H}=g\beta B\cdot \hat{S}+A\hat{S}\cdot \hat{I}+D({\hat{S}}_{z}^{2}-\frac{1}{3}S(S+1))$$


Here, *g* is the *g*-factor of the broader signal; *β* the Bohr magneton; *B* the static magnetic field; *S* and *I* the electron and nuclear spins of the paramagnetic center, respectively; and *A* and *D* the hyperfine constant and zero field splitting (ZFS) constant, respectively. Another ZFS constant, *E*, is neglected because change in *E* causes little impact to simulation of the Nd^3+^ signal. The simulated ESR signals of Nd^3+^ in BaTiO_3_ crystals and in the BN4T-2 ceramics are shown in Figure 5.

It is well known that the ESR of Nd^3+^ in BaTiO_3_ crystals can be observed only in the rhombohedral phase of BaTiO_3_ at low temperatures (e.g., *T* = 7 K) [16] and in other host crystals can also be detected at low temperatures (e.g., *T* ≤ 20 K) using the X-band frequency [25,26,27,28,29]. Neodymium has six stable isotopes; two of them, ^145^Nd^3+^ and ^143^Nd^3+^ (8.3 and 12.17% of nature abundance, *I* = 7/2), are odd isotopes with a non-zero magnetic moment. At low temperatures, Nd^3+^ in BaTiO_3_ [16] or in other hosts [25,26,27] creates 17 ESR lines: one strong line caused by all non-magnetic isotopes with *I* = 0 and two hyperfine octets by ^145^Nd^3+^ and ^143^Nd^3+^ [25,26,27], as shown in Figure 5a. For Nd-doped BaTiO_3_ crystals, this signal is comparatively weak; at *T* ≥ 10 K, the spectrum of Nd^3+^ vanishes [16]. The powder spectra of rare-earth ions (such as Eu^2+^, Gd^3+^, and Tb^4+^) in ceramics usually show a broader signal at RT [9,10,11,12]. It was reported that the ESR linewidth of Nd^3+^ ions at low temperatures generally increased from 1 Gs at *T* = 7 K to 350 Gs at *T* = 17 K with increasing temperature [25]. The ESR spectrum of BN4T-2 is simulated as the linewidth of the Nd^3+^ signal at 243 Gs, *A* = 15 MHz, and *D* = 85 MHz (Figure 5b). Obviously, the broader signal in BN4T-2 ceramic arises from a broadening of an intense line from isotopes with *I* = 0 in crystal, rather than from those weak hyperfine lines from ^145^Nd^3+^ and ^143^Nd^3+^. This simplifies the deciphering of spectra.

The powder spectra of Nd^3+^ at RT had never been reported. For single crystals with different host lattices, there are some differences in *g* value. Possenriede et al. and Falin et al. gave *g* = ~2.5 for Nd^3+^ in BaTiO_3_ [16] and a mean *g* value of 2.509 for Nd^3+^ in KZnO_3_ perovskite [28], respectively, which are very analogous to our *g* value (*g* = 2.338) in BN4T-2. In particular, Asatryan and Rosa gave the *g* tensors *g_x_* = 2.83, *g_y_* = 2.58, and *g_z_* = 1.69 at *f* = 9.24 GHz for Nd^3+^ in YAlO_3_ single crystals [28], and the mean *g* factor <*g*> = (*g_x_* + *g_y_* +*g_z_*)/3 = 2.37, which agrees quite well with the value of the *g* factor obtained for Nd^3+^ in BN4T-2. Thus, the broader signal at *g* = 2.338 should originate from a superposition effect of all *g* tensors of Nd^3+^ in the tetragonal BN4T-2 grains.

### 4.2. Probable Origin and Simulation of Nd^3+^ ESR in BN4TC5 at RT

The BN4TC5-2 sample exhibits a narrow *g* = 2.151 ESR signal with a linewidth of 26 Gs (Figure 4b). This signal can be observed in all ESR spectra of high-permittivity (Ba_1–*x*_Nd*_x_*)(Ti_1–*x*/4–*y*_Ce*_y_*)O_3_ (BNTC) ceramics prepared in 2006 and is attributed to Nd^3+^ in ${\mathrm{Nd}}_{\mathrm{Ba}}^{•}-{\mathrm{Ce}}_{\mathrm{Ti}}^{\times }$ complexes [18], because *g*_⊥_ = 2.16 of Nd^3+^ (3d^3^, ^4^I_9/2_) in the *g*-tensor [30] is close to the *g* value of this signal in BNTC.

Nd^3+^ in singly and doubly doped BaTiO_3_ ceramics exhibits two different kinds of ESR signals after long-term preservation of samples. Our experiments reveal that at a sintering temperature (*T*_s_) of 1450 °C, Ce is not easily incorporated as Ce^4+^ into the Ti-sites in the BaTiO_3_ lattice because of the relatively larger ionic size of Ce^4+^ (0.87 Å) compared to Ti^4+^ (0.605 Å) [31]. A higher sintering temperature, such as *T*_s_ = 1540 °C, is therefore required to obtain single-phase Ba(Ti_1–*y*_Ce*_y_*)O_3_ ceramics [32]. The single-phase BN4TC5 can be formed at a lower *T*_s_ = 1450 °C due to the formation of ${\mathrm{Nd}}_{\mathrm{Ba}}^{•}-{\mathrm{Ce}}_{\mathrm{Ti}}^{\times }$ defect complexes [18]. The simulation of the ESR of BN4TC5 indicates that the linewidth of the Nd^3+^ signal is 32 Gs, *A* = 15 MHz, and *D* = 110 MHz (Figure 5c). There is a great difference in the ZFS constant *D* between the pseudo-cubic BN4TC5 and the tetragonal BN4T. The ${\mathrm{Nd}}_{\mathrm{Ba}}^{•}-{\mathrm{Ce}}_{\mathrm{Ti}}^{\times }$ complexes and an expanded lattice in BN4TC5 compared to BN4T may be responsible for the narrow *g* = 2.151 signal of Nd^3+^ in BN4TC5. Thus, these two kinds of RT-accessible ESR signals can be attributed to the point defects ${\mathrm{Nd}}_{\mathrm{Ba}}^{•}$ in singly doped BaTiO_3_ and the defect complexes ${\mathrm{Nd}}_{\mathrm{Ba}}^{•}-{\mathrm{Ce}}_{\mathrm{Ti}}^{\times }$ in doubly doped BaTiO_3_, respectively.

### 4.3. Rhombohedral Distortions Detected by Raman Scattering and Photoluminescence

To further investigate the origin of the ESR signals, the Raman spectra of BN4T and BN4TC5 were re-measured in 2017. Similar to Er-doped BaTiO_3_ ceramics [14], the Raman scattering spectra and photoluminescence (PL) of Nd^3+^ can occur simultaneously as two distinct optical processes upon 532 nm excitation for BN4T and BN4TC5 ceramic powders, as shown in Figure 6.

BN4T and BN4TC5 exhibit four common phonon bands in the tetragonal BaTiO_3_, with peaks at 265 [A_1_ (TO_2_)], ~303 [B_1_ + E], ~520 [A_1_ (TO_3_)], and ~720 cm^−1^ [A_1_ (LO_3_) + E (LO_3_)] (Figure 6 insets). A band at ~834 cm^−1^ is caused by the occupations of Ba^2+^ sites by Nd^3+^ in BN4T and BN4TC5 [3]. The presence of the peak at 300 cm^−1^ reveals the pseudo-cubic nature of BN4TC5 containing the ferroelectric phases. In BaTiO_3_ ceramics, several bands at ~171, ~186, ~243, ~305, and ~480 cm^−1^ [E (TO_4_)] are characteristic of low-temperature rhombohedral and orthorhombic phases, and their intensity increases with decreasing temperature [33,34,35]. Four weak bands at 167–172, 187–189, 244–248, and 302–304, and strong peaks at 478–810 cm^−1^ were observed for the aged BN4T and BN4TC5 powders, which implies that the rhombohedral distortions, which cannot be detected by XRD at RT (Figure 1), exist in these two ceramics preserved over a long period of time. This may be a reason why two Nd^3+^-related signals can be detected at RT.

Two strong PL bands at 804 and 876 nm in the spectrum of BN4T and BN4TC5 originate from the ^4^F_5/2_→^4^I_9/2_ and ^4^F_3/2_→^4^I_9/2_ transitions of Nd^3+^ [8,36,37,38]. BN4TC5 exhibits enhanced photoluminescence relative to BN4T. If the Nd^3+^ ion experiences a crystal field of cubic symmetry, the fourfold degeneracy of the ^4^F_3/2_ level is not split. Therefore, in the cubic hosts, any splitting of the ^4^F_3/2_ level is indicative of association of the Nd impurity with other defects [39]. BN4TC5 has a pseudo-cubic structure and some Nd^3+^ ions located in cubic symmetry. A clear peak at 896 nm can be observed for the ^4^F_3/2_→^4^I_9/2_ transition of Nd^3+^ in BN4TC5, but it is absent for the tetragonal BN4T. This fact further reveals that Nd^3+^ ions in BN4T and BN4TC5 form two different types of Nd^3+^ centers, i.e., ${\mathrm{Nd}}_{\mathrm{Ba}}^{•}$ in singly doped BaTiO_3_ and the defect complexes ${\mathrm{Nd}}_{\mathrm{Ba}}^{•}-{\mathrm{Ce}}_{\mathrm{Ti}}^{\times }$ in doubly doped BaTiO_3_, which are responsible for the broader *g* = 2.338 and narrow *g* = 2.151 signals, respectively. The association of ${\mathrm{Nd}}_{\mathrm{Ba}}^{•}$ with $V_{\mathrm{Ti}}^{\prime \prime \prime \prime }$ can be neglected because Ti vacancies are present in both BN4T and BN4TC5.

### 4.4. Controversy over Probable Origin of Nd^3+^ ESR in BN4T and BN4TC5

It is known that the ESR of Nd^3+^ in BaTiO_3_ crystals can be observed only in the rhombohedral phase at low temperatures, and at *T* ≥ 10 K, the spectrum vanishes [16]. Although the weaker rhombohedral distortions exist in the aged BN4T and BN4TC5 powders (Figure 6), the two signals at *g* = 2.151 and 2.338 should have nothing to do with the rhombohedral phase because of the short spin-lattice relaxation time of Nd^3+^ at RT. There is much controversy over the abovementioned origin of Nd^3+^ ESR in BN4T and BN4TC5.

The ESR of Pr^4+^ in BaTiO_3_ will facilitate the understanding of the probable origin of Nd^3+^ ESR in BN4T and BN4TC5. Pr^4+^ (4*f*^1^) has the same short spin-lattice relaxation time as Nd^3+^ (4*f*^1^) because the ESR of Pr^4+^ in BaCeO_3_ can be observed only at a very low temperature of 8.5 K [40]. However, a thermally accessible *g* = 3.77, ESR signal associated with Ti-site Pr^4+^ was unexpectedly observed above 120 °C, which relates to the preservation history of the Pr-doped BaTiO_3_ sample [13]. This ESR signal is well resolved and characteristic of multiple lines of Pr^4+^. This example seems to imply a change in the spin-lattice relaxation time with the preservation history of samples, though it is in conflict with the fundamental principle of the short spin-lattice relaxation time of Pr^4+^ at RT.

The Raman scattering investigations reveal that for the aged BN4T and BN4TC5 powders which were heat-treated, several Raman bands associated with the rhombohedral distortions vanish (Figure 6). Correspondingly, the two ESR signals at *g* = 2.151 and 2.338 vanished (Figure 3 and Figure 4). Thus, it is undeniable that the rhombohedral distortions in the aged BN4T and BN4TC5 powders may be a reason why two Nd^3+^-related signals can be detected at RT. However, it is still an unsolved mystery as to why the spin-lattice relaxation time changes in the BN4T and BN4TC5 powders stored for 11 years.

### 4.5. Aging-Resistant Behavior of BN4TC5

After being stored for 11 years, BN4TC5 shows a decrease in *ε*′_m_ by 6% and a slower shift in *T*_m_ from 35 °C in 2006 to 40 °C in 2017 (Figure 2a). The slightly expanded lattice (Figure 1) and rhombohedral distortion formed by the long-term preservation (Figure 6) are responsible for the decrease in *ε*′_m_ (Figure 2a). This is because on the basis of the *ε*′–*T* curve of BaTiO_3_, both lattice expansion in higher temperatures and rhombohedral distortion in lower temperatures give rise to a decrease in *ε*′. The time-induced *T*_m_-shift to higher temperature seems to be an universal law for co-doped BaTiO_3_ ceramics, e.g., *T*_m_ on heating was observed to shift from 36 °C in 2004 [41] to 38 °C in 2006 [42] for the high-permittivity (Ba_0.97_La_0.03_)(Ti_0.9425_Ce_0.05_)O_3_ ceramic. The slightly expanded lattice makes the main contribution to this *T*_m_-shift.

On the basis of the time-induced decrease in the ESR intensity of Mn^2+^ (Figure 4), it is inferred that Mn^2+^ impurities due to long-term preservation can be gradually oxidized to higher-valence-state Mn^4+^/Mn^3+^, releasing a certain number of electrons. This is the main reason for time-induced increase in tan *δ* above RT (Figure 2b). Although the *T*_m_-shift results in slight invalidation of the Y5V specification (Figure 2a), BN4TC5 is considered to be a better aging-resistant dielectric.

## 5. Conclusions

After they had been preserved over a long period of time, (Ba_0.96_Nd_0.04_)Ti_0.99_O_3_ (BN4T) with a tetragonal structure and (Ba_0.96_Nd_0.04_)(Ti_0.94_Ce_0.05_)O_3_ (BN4TC5) ceramics with a pseudo-cubic structure exhibited a broader ESR signal at *g* = 2.338 and a narrow ESR signal at *g* = 2.151 at RT, respectively, which are most likely originated from Nd^3+^ Kramers ions in BN4T and a Nd^3+^–Ce^4+^ defect complex in BN4TC5. The Raman scattering investigations provided evidence that some rhombohedral distortions in ceramics, which cannot be detected by XRD at room temperature, formed owing to long-term preservation of the two samples. Thermal treatment gave rise to the disappearance of both the rhombohedral distortions and Nd^3+^-related signals. The reason for the ESR of Nd^3+^ in the aged BN4T and BN4TC5 powders at room temperature needs to be further studied. This finding will have far-reaching implications for RT-accessible realization of low-temperature detection methods. For BN4TC5 stored for 11 years, a slight decrease in *ε*′_m_ and a slower shift in *T*_m_ relate to the slightly expanded lattice and its rhombohedral distortion caused by long-term preservation. The oxidation of Mn impurities with time is responsible for a slight increase in tan *δ* at RT. As a whole, BN4TC5 is considered to be a better aging-resistant dielectric.

## Figures and Tables

**Figure 1 materials-12-00451-f001:**
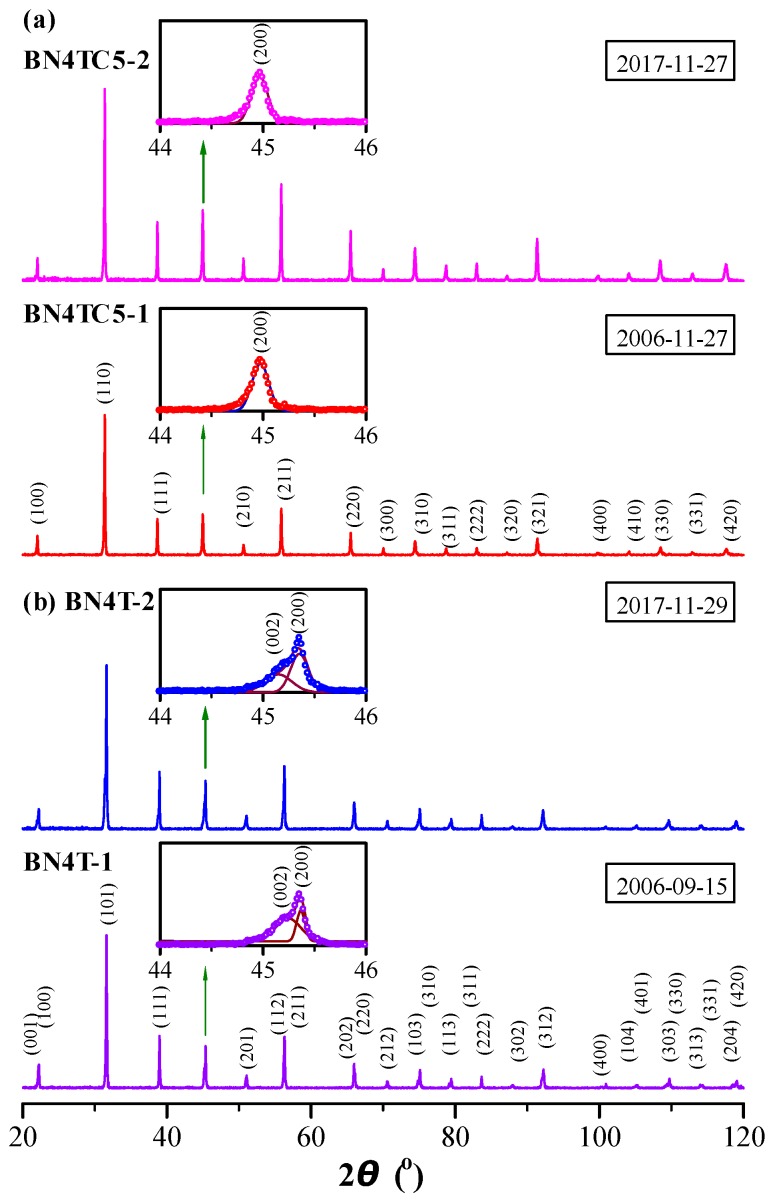
Powder XRD patterns of (**a**) BN4TC5 and (**b**) BN4T ceramics, measured in 2006 and again in 2017. The four insets on the left depict Gaussian fitting of the XRD peaks in the vicinity of 45°.

**Figure 2 materials-12-00451-f002:**
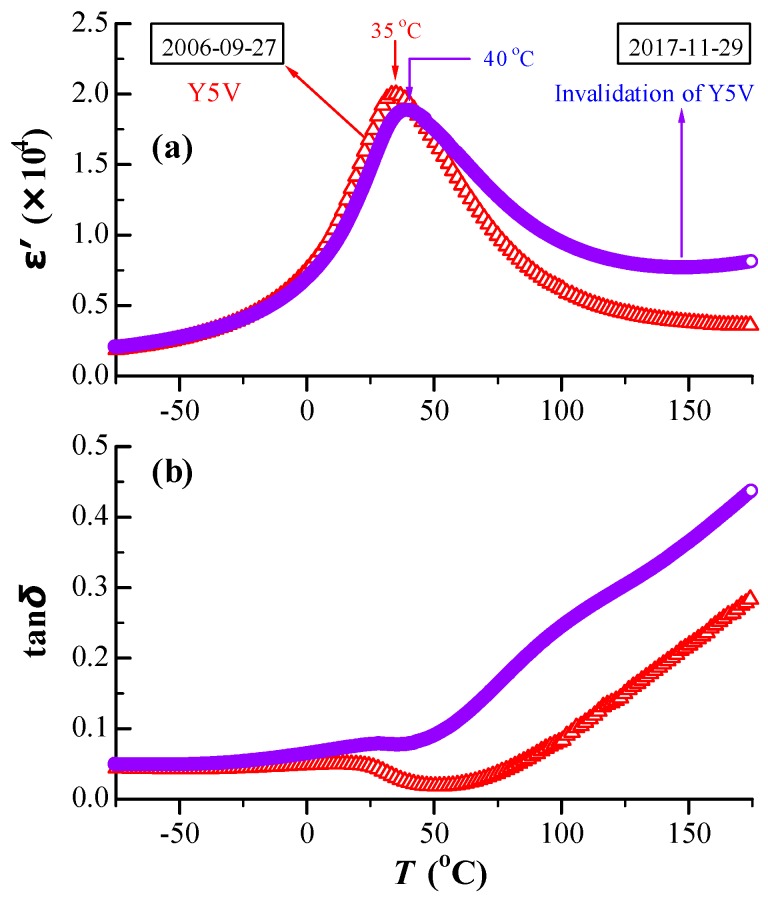
Temperature dependences of (**a**) *ε*′ and (**b**) tan*δ* for BN4TC5, measured in 2006 and again in 2017.

**Figure 3 materials-12-00451-f003:**
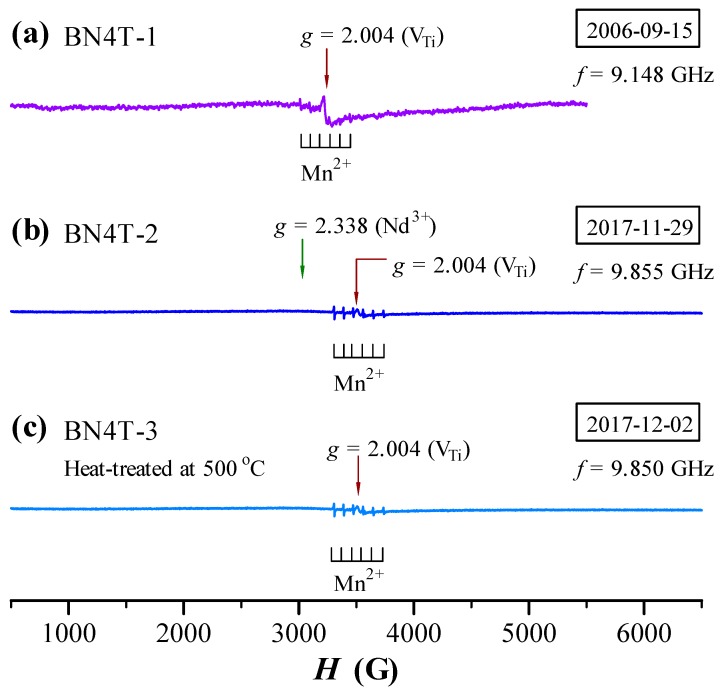
ESR spectra of BN4T in Table 1, measured (**a**) in 2006 using a JES-RE3X spectrometer (JEOL) and (**b**,**c**) in 2017 using an A300-10/12 spectrometer (Bruker, Billerica, MA, USA).

**Figure 4 materials-12-00451-f004:**
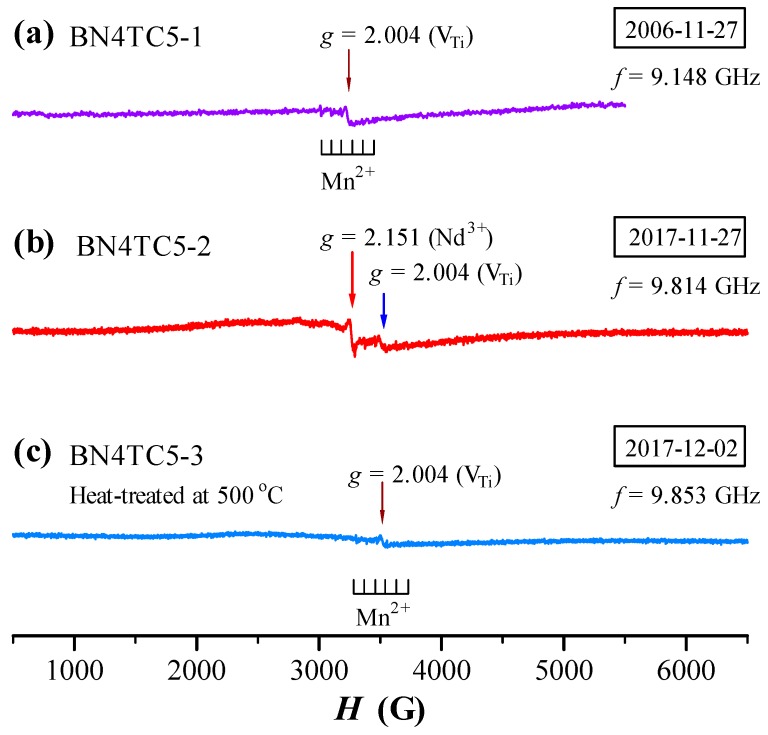
ESR spectra of BN4TC5, measured in (**a**) 2006 using a JES-RE3X spectrometer (JEOL) and in (**b**,**c**) 2017 using an A300-10/12 spectrometer (Bruker, Billerica, MA, USA).

**Figure 5 materials-12-00451-f005:**
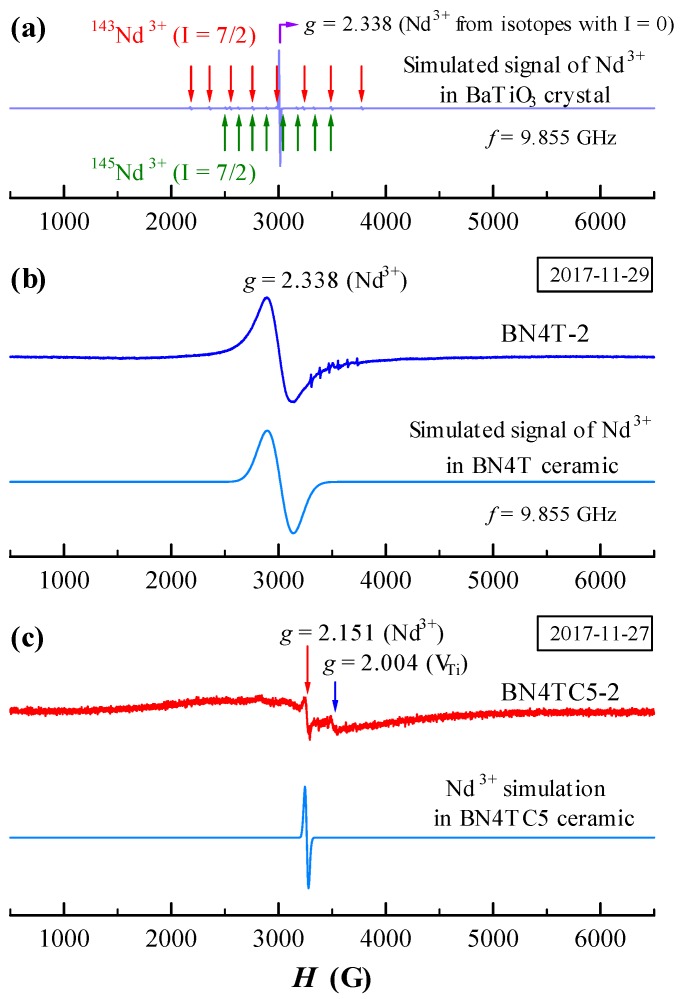
(**a**) Simulated ESR signal for Nd^3+^ in BaTiO_3_ crystal; ESR signals observed and simulated for Nd^3+^ in (**b**) BN4T-2 and (**c**) BN4TC5-2.

**Figure 6 materials-12-00451-f006:**
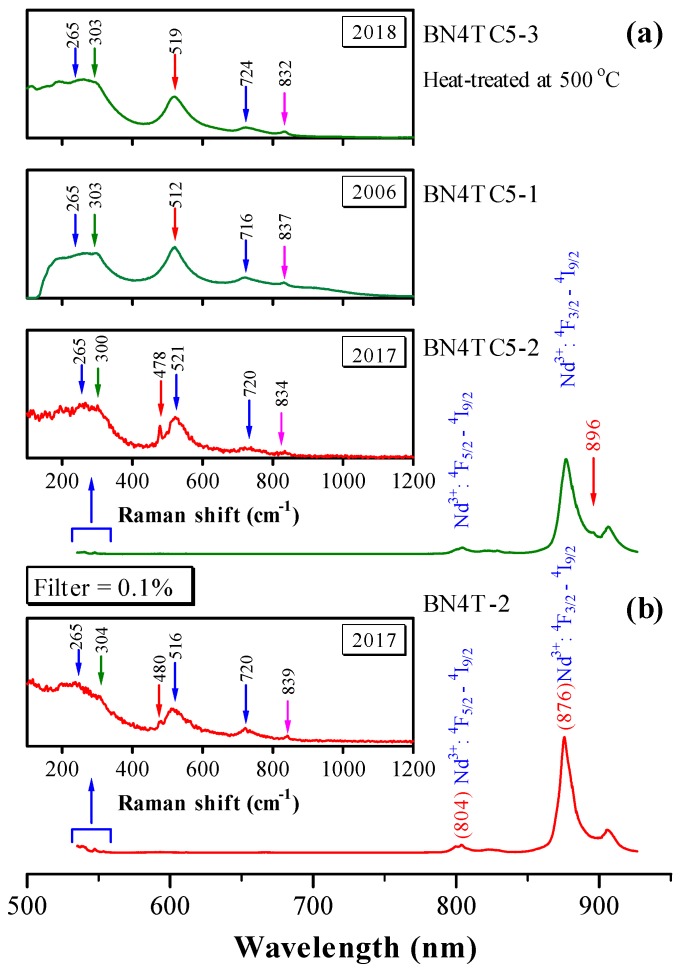
Raman scattering spectra and photoluminescence of (**a**) BN4TC5 and (**b**) BN4T, shown in wavelength. Excitation: 532 nm laser line. The insets show enlarged pure Raman signals at around 550 nm, shown in Raman shift.

**Table 1 materials-12-00451-t001:** Symbols and preservation history of BN4T and BN4TC5.

Symbol	Date of Preparation	Date of ESR	Preservation History
BN4T-1	9 September 2006	15 September 2006	Ceramic powder in a desiccator
BN4T-2	9 September 2006	29 November 2017	Ceramic powder in a desiccator
BN4T-3	9 September 2006	2 December 2017	Ceramic powder was heat treated at 500 °C for 2 h
BN4TC5-1	10 October 2006	27 November 2006	Ceramic powder in a desiccator
BN4TC5-2	10 October 2006	27 November 2017	Ceramic powder in a desiccator
BN4TC5-3	10 October 2006	23 August 2018	Ceramic powder was heat treated at 500 °C for 2 h

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
