# Peer review of "Aging-Resistant Behavior and Room Temperature Electron Spin Resonance of Nd3+ in Singly and Doubly Doped BaTiO3 Ceramics Associated with Preservation History"

_materials, 2019, doi:10.3390/ma12030451_

Round 1

Reviewer 1 Report

This manuscript entitled Aging resistant behavior and room-temperature electron spin resonance of Nd3+ in singly- and doubly-doped BaTiO3 ceramics associated with preservation history is interesting and well written.  Particular dielectric and aging resistance behavior and presented results interesting and useful for Y5V specifications. I recommend this paper to publish in Materials journal.

Author Response

Response to Reviewer 1 Comments

Point 1: Open Review: I don't feel qualified to judge about the English language and style.

Response 1: We have asked for MDPI to provide an English editing service on the revised manuscript.

Reviewer 2 Report

The paper entitled “Aging resistant behavior and room-temperature electron spin resonance of Nd3+ in singly- and doubly-doped BaTiO3 ceramics associated with preservation history” by Lu et al. deals with the aging resistant behavior and room-temperature electron spin resonance of doped ferroelectrics. The study is rather detailed, but there are still some issues that prevent publishing the manuscript in the present form.  The authors must respond to the following comments/questions: 

1.      In introduction authors are suggested to clearly highlight the advantages of this work with suitable references.

2.      The sentence “The well-mixed mixtures were calcined in air at 1100 °C for 5 h to decarbonate.” is not clear. What does decarbonate means? In calcinations many things happen.

3.      Why the dielectric permittivity increases and ferroelectric Tc decreases of the sample prepared in 2017 compared to 2006. There should be more discussion on this.

4.      Author should explain the importance of the work done from the application point of view.

5.      Authors must include some recent publications in reference list.

6.      There are many sentences with grammar or typo problems; a careful reading is required.

Author Response

Response to Reviewer 2 Comments

Point 1: In introduction authors are suggested to clearly highlight the advantages of this work with suitable references.

Response 1: In the revised manuscript, please see Page 2, lines 53‒62:

Ferroelectric ceramics are subject to degradation either during electrical loading (fatigue) or with time in the absence of an external mechanical or electrical load (aging) [19]. Recent studies on aging behavior mainly focused on mechanisms of piezoelectric ceramics [19,20] and change in polarization‒electric field (PE) hysteresis loops [15]. However, the time-induced aging effect has prompted surprisingly little concern in the dielectric behaviour. The high-permittivity (Ba1−xNdx)(Ti1−x/4−yCey)O3 (x = 0.04) ceramic meets the Y5V specification (−82 % ≤ (ε'−ε'RT)/ε'RT ≤ +22 % in the temperature range −30 to 85 °C) [18], making it ideal for decoupling applications within a limited temperature range. The investigation of the time-induced aging property of this ceramic and report on its advantageous aging-resistant performance (a decrease in ε'm by 6 % after 12 years) are of great importance in the actual Y5V application.

Point 2: The sentence “The well-mixed mixtures were calcined in air at 1100 °C for 5 h to decarbonate.” is not clear. What does decarbonate means? In calcinations many things happen.

Response 2: In the revised manuscript, please see Page 2, lines 65‒71:

The initial materials were reagent-grade BaCO3, TiO2, CeO2 (Kanto Chem., Tokyo, Japan) and Nd2O3 (Wako Pure Chem., Tokyo, Japan). (Ba0.96Nd0.04)Ti0.99O3 (BN4T) and (Ba0.96Nd0.04)(Ti0.94Ce0.05)O3 (BN4TC5) ceramics were prepared in 2006 using the mixed oxide route, as described elsewhere [10]. The stoichiometric mixtures in accordance with the above metal ratios (Ba/Nd/Ti) for BN4T and (Ba/Nd/Ti/Ce) for BN4TC5 were carefully mixed. The mixtures were calcined in air at 1100 °C for 5 h to decarbonate the initial material BaCO3. The purpose of “decarbonating” in the calcination stage is to avoid the release of CO2 and to benefit ceramic densification in the following sintering stage.

Point 3: Why the dielectric permittivity increases and ferroelectric TC decreases of the sample prepared in 2017 compared to 2006. There should be more discussion on this.

Response 3: In this work, we only focused on the aging-resistant behavior of BN4TC5 rather than BN4T because BN4TC5 is a valuable Y5V dielectric. The BN4TC5 sample with specification was prepared in 2006. We did not re-prepare this sample in 2017. To observe its dielectric aging behavior, we measured temperature dependences of ε' and tan δ for BN4TC5 in 2006 and 2017, respectively, as shown in Fig. 2. Please see Page 2, lines 66‒67 and Fig. 2 caption:

(Ba0.96Nd0.04)Ti0.99O3 (BN4T) and (Ba0.96Nd0.04)(Ti0.94Ce0.05) O3 (BN4TC5) ceramics were prepared in 2006 using the mixed oxide route…

Fig. 2.  Temperature dependences of (a) ε' and (b) tan δ for BN4TC5, measured in 2006 and again in 2017.

Please see Page 9, line 265 ‒ Page 10, line 279:

After being stored for 11 years, BN4TC5 shows a decrease in ε'm by 6 % and a slower shift in Tm from 35 °C in 2006 to 40 °C in 2017 (Figure 2a). The slightly expanded lattice (Figure 1) and rhombohedral distortion formed by the long-term preservation (Figure 6) are responsible for the decrease in ε'm (Figure 2a). This is because on the basis of the ε'‒T curve of BaTiO3, both lattice expansion in higher temperatures and rhombohedral distortion in lower temperatures give rise to a decrease in ε'. The time-induced Tm-shift to higher temperature seems to be an universal law for co-doped BaTiO3 ceramics, e.g. Tm on heating was observed to shift from 36 °C in 2004 [40] to 38 °C in 2006 [41] for the high-permittivity (Ba0.97La0.03)(Ti0.9425Ce0.05)O3 ceramic. The slightly expanded lattice makes the main contribution to this Tm-shift.

On the basis of the time-induced decrease in the ESR intensity of Mn2+ (Figure 4), it is inferred that Mn2+ impurities due to long-term preservation can be gradually oxidized to higher-valence-state Mn4+/Mn3+, releasing a certain number of electrons. This is the main reason for time-induced increase in tan δ above RT (Figure 2b). Although the Tm-shift results in slight invalidation of the Y5V specification (Figure 2a), BN4TC5 is considered to be a better aging-resistant dielectric.

Point 4: Author should explain the importance of the work done from the application point of view.

Response 4: In the revised manuscript, please see Page 2, lines 53‒62:

Ferroelectric ceramics are subject to degradation either during electrical loading (fatigue) or with time in the absence of an external mechanical or electrical load (aging) [19]. Recent studies on aging behavior mainly focused on mechanisms of piezoelectric ceramics [19,20] and change in polarization‒electric field (PE) hysteresis loops [15]. However, the time-induced aging effect has prompted surprisingly little concern in the dielectric behaviour. The high-permittivity (Ba1−xNdx)(Ti1−x/4−yCey)O3 (x = 0.04) ceramic meets the Y5V specification (−82 % ≤ (ε'−ε'RT)/ε'RT ≤ +22 % in the temperature range −30 to 85 °C) [18], making it ideal for decoupling applications within a limited temperature range. The investigation of the time-induced aging property of this ceramic and report on its advantageous aging-resistant performance (a decrease in ε'm by 6 % after 12 years) are of great importance in the actual Y5V application.

Point 5: Authors must include some recent publications in reference list.

Response 5: The following four older references in the first manuscript were removed from the revised manuscript.

15.  Kröger, F.A.; Vink, H.J. Solid State Physics III. New York: Academic Press. 1956.

30. Buscaglia, V.; Buscaglia, M.T.; Viviani, M.; Ostapchuk, T.; Gregora, I.; Petzelt, J.; Mitoseriu, L.; Nanni, P.;  Testino, A.; Calderone, R.; Harnagea, C.; Zhao, Z.; Nygren, M. Raman and AFM piezoresponse study of dense BaTiO3 nanocrystalline ceramics. J. Eur. Ceram. Soc. 2005, 25, 3059–3062.

31. Buscaglia, V.; Buscaglia, M.T.; Viviani, M.; Mitoseriu, L.; Panni, P.; Trefiletti, V.; Piaggio, P.; Gregora, I.;  Ostapchuk, T.; Pokorny, J.; Petzelt, J. Raman and AFM piezoresponse study of dense BaTiO3 nanocrystalline ceramics. J. Eur. Ceram. Soc. 2006, 26, 2889–2898.

32. Shiratori, Y.; Pithan, C.; Dornseiffer, J.; Waser, R. Raman scattering studies on nanocrystalline BaTiO3 Part – consolidated polycrystalline ceramics. J. Raman. Spectrosc. 2007, 38, 1300–1306.

In the revised manuscript, the eight recent publications associated with aging behaviour, EPR and Raman spectroscopy were given in reference list, as follows:

15. Alkathy, M.S.; James Raju, K.C. Onset of multiferroicity in nickel and lithium co-substituted barium titanate ceramics. J. Magn. Magn. Mater. 2018, 452, 40–47.

17. Eremina, R.; Gavrilova, T.; Yatsyk, I.; Fazlizhanov, I.; Likerov, R.; Shustov, V.; Zavartsev, Y.; Zagumennyi, A.; Kutovoi, S. Investigations of Y2SiO5: Nd143 by ESR method. J. Magn. Magn. Mater. 2017, 440, 13–14.

19. Genenko, Y.A.; Glaum, J.; Hoffmann, M.J.; Albe, K. Mechanisms of aging and fatigue in ferroelectrics. Mater. Sci. Eng. B 2015, 192, 52–82.

20. Fan, Z.; Tan, X. In-situ TEM study of the aging micromechanisms in a BaTiO3-based lead-free piezoelectric ceramic. J. Eur. Ceram. Soc. 2018, 38, 3472–3477.

21. Zhao, X.; Chen, W.; Zhang, L.; Zhong, L. The effect of the bipolar field on the aging behavior and the associated properties of the Mn-doped BaTiO3 ceramics. J. Alloy. Compd. 2015, 618, 441–445.

32. Hayashi, H.; Nakamura, T.; Ebina T. In-situ Raman spectroscopy of BaTiO3 particles for tetragonal–cubic transformation. J. Phys. Chem. Solids 2013, 74, 957–962.

33. Lin, Y.T.; Ou, S.F.; Lin, M.H.; Song, Y.R. Effect of MgO addition on the microstructure and dielectric properties of BaTiO3 ceramics. Ceram. Int. 2018, 44, 3531–3535.

34. Liu, Q.; Liu, J.; Lu, D.; Zheng, W. Colossal dielectric behavior and relaxation in Nd-doped BaTiO3 at low temperature. Ceram. Int. 2018, 44, 7251–7258.

Point 6: There are many sentences with grammar or typo problems; a careful reading is required.

Response 6: We have asked for MDPI to provide an English editing service on the revised manuscript.

Reviewer 3 Report

-In the reference 17 does not show the preparation of ceramics. - Attention to editing (for example: as described elsewhere. [17])-  I suggest the manuscript checked thoroughly and corrected.

Author Response

Open Review: I don't feel qualified to judge about the English language and style.

Response: We have asked for MDPI to provide an English editing service on the revised manuscript.

Point 1: In the reference 17 does not show the preparation of ceramics. - Attention to editing (for example: as described elsewhere. [17])-  I suggest the manuscript checked thoroughly and corrected.

Response 1: We are sorry for the reference number mistake for our carelessness. The cited reference should be Ref. 10. In the revised manuscript, please see Page 2, lines 610:

 The initial materials were reagent-grade BaCO3, TiO2, CeO2 (Kanto Chem.) and Nd2O3 (Wako Pure Chem.). (Ba0.96Nd0.04)Ti0.99O3 (BN4T) and (Ba0.96Nd0.04)(Ti0.94Ce0.05) O3 (BN4TC5) ceramics were prepared in 2006 using the mixed oxide route, as described elsewhere [10]. The stoichiometric mixtures in accordance with the above metal ratios (Ba:Nd:Ti) for BN4T and (Ba:Nd:Ti:Ce) for BN4TC5 were carefully mixed. The mixtures were calcined in air at 1100 °C for 5 h to decarbonate the initial material BaCO3.

Round 2

Reviewer 3 Report

I have no comments.